# Heart failure medication after a first hospital admission and risk of heart failure readmission, focus on beta-blockers and renin-angiotensin-aldosterone system medication: A retrospective cohort study in linked databases

Willemien J. Kruik-Kollöffel[1]*, Job van der Palen[2,3], Carine J. M. Doggen[4], Marissa C. van Maaren[4,5], H. Joost Kruik[6], Edith M. Heintjes[7], Kris L. L. Movig[8], Gerard C. M. Linssen[6]

1 Department of Clinical Pharmacy, Ziekenhuisgroep Twente (Hospital Group Twente), Almelo and Hengelo, the Netherlands, 2 Medical School Twente, Medisch Spectrum Twente, Enschede, the Netherlands, 3 Department of Research Methodology, Measurement and Data Analysis, University of Twente, Enschede, the Netherlands, 4 Department of Health Technology and Services Research, Technical Medical Centre, University of Twente, Enschede, the Netherlands, 5 Department of Research and Development, Netherlands Comprehensive Cancer Organisation (IKNL), Utrecht, the Netherlands, 6 Department of Cardiology, Ziekenhuisgroep Twente (Hospital Group Twente), Almelo and Hengelo, the Netherlands, 7 PHARMO Institute, Utrecht, the Netherlands, 8 Department of Clinical Pharmacy, Medisch Spectrum Twente, Enschede, the Netherlands

* w.kruik@sxb.nl

## Abstract

### Background

This study assessed the association between heart failure (HF) medication (angiotensin-converting-enzyme inhibitors (ACEI)/angiotensin-receptor blockers (ARB), beta-blockers (BB), mineralocorticoid-receptor antagonists (MRA) and diuretics) and HF readmissions in a real-world unselected group of patients after a first hospital admission for HF. Furthermore we analysed readmission rates for ACEI versus ARB and for carvedilol versus β1-selective BB and we investigated the effect of HF medication in relation to time since discharge.

### Methods and findings

Medication at discharge was determined with dispensing data from the Dutch PHARMO Database Network including 22,476 patients with HF between 2001 and 2015. After adjustment for age, gender, number of medications and year of admission no associations were found for users versus non-users of ACEI/ARB (hazard ratio, HR = 1.01; 95%CI 0.96–1.06), BB (HR = 1.00; 95%CI 0.95–1.05) and readmissions. The risk of readmission for patients prescribed MRA (HR = 1.11; 95%CI 1.05–1.16) or diuretics (HR = 1.17; 95%CI 1.09–1.25) was higher than for non-users. The HR for ARB relative to ACEI was 1.04 (95%CI 0.97–1.12) and for carvedilol relative to β1-selective BB 1.33 (95%CI 1.20–1.46). Post-hoc

**Data Availability Statement:** Original de-identified patient data used in this analysis were obtained from the PHARMO Database Network. The PHARMO Institute provided data to Medisch Spectrum Twente/Ziekenhuisgroep Twente, which were used to create the analytical files for the study. Medisch Spectrum Twente/Ziekenhuisgroep Twente does not have the right to provide these files to a third party. Data from this study are therefore only available upon reasonable request, for researchers who meet the criteria for access to confidential data on a patient level. The corresponding author should be informed and a formal request should be addressed to the PHARMO Institute (pharmo@pharmo.nl). Requests will be submitted to the STIZON Compliance Committee for approval, which will have to judge if the person that requests the data will be eligible to access the data.

**Funding:** The author(s) received no specific funding for this work.

**Competing interests:** The authors have declared that no competing interests exist.

analyses showed a protective effect shortly after discharge for most medications. For example one month post discharge the HR for ACEI/ARB was 0.77 (95%CI 0.69–0.86). Although we did try to adjust for confounding by indication, probably residual confounding is still present.

## Conclusions

Patients who were prescribed carvedilol have a higher or at least a similar risk of HF readmission compared to β1-selective BB. This study showed that all groups of HF medication -some more pronounced than others- were more effective immediately following discharge.

## Introduction

The evidence-based medical treatment of heart failure (HF) with reduced left ventricular function (HFrEF) includes angiotensin-converting-enzyme inhibitors (ACEI) or angiotensin-receptor blockers (ARB) as an alternative for patients who do not tolerate an ACEI, beta-blockers (BB), and mineralocorticoid-receptor antagonists (MRA). With diuretics, these groups of medications form the foundation of the treatment of HF. Over the years, not only the treatment of HF has dramatically changed, the type of patients diagnosed with HF has changed as well [1]. Patients are older and have more comorbidities. Due to more use of therapies to prevent the onset of HF and the expanding treatment options for HF, they live more years at risk for hospital admissions.

The evidence for the use of the main groups of HF medication (ACEI/ARB, BB, MRA and diuretics) is based upon large randomized clinical trials (RCTs) published mainly in the years 1999–2003. However, trial populations do not resemble daily clinical practice of HF patients [2]. It is therefore essential to re-establish the effectiveness of HF medication in a real-world setting. Hospital readmission for HF after a first admission is a widely used indicator to investigate this as patients have a high risk of HF readmission [3–8]. As the risk of readmission and mortality is highest early after hospital discharge, prescription and dose-titration of disease-modifying medications (ACEI/ARB, BB and MRA) especially in patients with HFrEF should start during hospital admission [9, 10].

The aim of this study was to assess over a 15-year period the association between HF medication prescribed at discharge (ACEI/ARB, BB, MRA and diuretics) and HF readmissions in a real-world, large, unselected group of patients after a first hospital admission for HF. We were particularly interested in differences in readmission rates for ACEI versus ARB and for β1-selective BB (sBBHF: bisoprolol, metoprolol and nebivolol) versus carvedilol, a β- and partly α1-blocking agent, as these medications are to some extent interchangeable.

## Methods

### Patient population and medication

In a cohort of 22,476 patients the association between HF medication at discharge after a first hospital admission for HF between January 1, 2001 and December 31, 2015 and readmission rates were analysed. All data were extracted from the PHARMO Database Network, a population-based, medical record linkage system covering more than four million Dutch inhabitants at the time of the study [11]. Its linkage algorithms have been validated and the Database Network forms a representative sample of the Dutch population [12].

Patients with a primary discharge diagnosis of HF (ICD-9 428; ICD-10 I50) or hypertensive heart disease with (congestive) HF (ICD-9 402; ICD-10 I11.0) were included. It was considered to be the first admission for HF if there was no known previous admission in at least 3 years, assuming one expects a patient in the Dutch health care system to be admitted to the same hospital with a rehospitalization for HF. Based on periods of uninterrupted data availability for both pharmacy and hospitalization data around the time of the hospitalization, for each patient periods of uninterrupted use were formed prior to analysis. Only patients with drug dispenses before as well as after admission were included, including at least one cardiovascular drug (ATC group C, cardiovascular system) at discharge. Data includes cardiovascular as well as non-cardiovascular medication, age at time of first hospital admission, gender, the year of admission and time to readmission or end of follow-up. The non-cardiovascular medication provides proxy information about co-morbidities. Last date of follow-up for readmissions was December 31, 2015. More details were previously reported [13].

## Data processing

Dispensing data were used as a proxy variable for medication use. Medication dispenses from primary and secondary care prescriptions, dispensed by out-patient pharmacies, were included. Dispenses from 4 months before HF hospital admission until 4 months after discharge were collected to determine the medication profile at discharge. Based on the last dispensing of a drug before hospital admission and the first dispensing after discharge, the medication profile on discharge was established, accepting a 30-day gap between consecutive dispensings as uninterrupted use of a specific class of drugs. A new drug started at discharge should be dispensed between 1 day before and 7 days after discharge to be assigned to the medication profile on discharge. Medication dispenses are coded according to the WHO Anatomical Therapeutic Chemical (ATC) Classification System. BB included were the ones recommended in the HF guidelines in Europe and the US: β1-selective BB (sBBHF: bisoprolol (C07AB07), metoprolol (C07AB02) and nebivolol (C07AB12)) and carvedilol (C07AG02), a β- and partly α1-blocking agent [3, 14]. Furthermore, ACEI (C09A and C09B), ARB (C09C and C09D), MRAs (C09DA) and diuretics (mainly loop diuretics, C03CA [13]) were investigated. For the main groups of heart failure medication the number of medications is calculated excluding the medication under surveillance. Patients using a certain medication are compared to all non-users, irrespective of what other HF medication was prescribed.

## Within-class comparison

ACEI and ARB are to some extent interchangeable, as well as sBBHF and carvedilol. A within-class analysis was performed between patients using either ACEI or ARB and between patients using sBBHF or carvedilol, irrespective of other HF medication.

## Post-hoc analysis on time since discharge

From previous research using the same database [15] we expected the results to be independent of the time since discharge. In a post-hoc analysis, however, we additionally analysed the data at time of follow-up at 1, 6, 12 and 24 months.

## Statistical analyses

Descriptive statistics were used to describe the cohort and to compare patients with and without specific HF medication. Data are presented as means (standard deviation, SD) or medians (interquartile range, IQR) for continuous variables and frequencies (%) for categorical

variables. Differences in continuous variables were compared using the *t*-test for independent samples or the Mann-Whitney U test, as appropriate. Between-group differences in categorical variables were compared using the Chi-squared test. The crude differences in readmission rates between the various medications versus non-use were visualized in the Kaplan–Meier curves and tested with the log-rank test. Patients without a readmission were censored at the end of follow-up. To optimize clarity of the Figures, censored patients (i.e. those who reached end of follow-up without readmission) are not shown. Subsequently a multivariable Cox regression model for each HF medication versus non-use was constructed to calculate the hazard ratio (HR) with 95% confidence intervals (CI). The Cox models were adjusted for baseline characteristics, i.e. age, gender, total number of unique medications and year of hospital admission. The HR and 95% CI for all medications are graphically displayed in a Forest-plot. A *P*-value of ≤0.05 was considered to indicate statistical significance. The statistical analysis for the within-class comparison was performed in the same way, comparing ACEI versus ARB and carvedilol versus sBBHF. Statistical analysis was performed using SPSS software version 24 (IBM SPSS Statistics, Armonk, New York, USA). The REporting of studies Conducted using Observational Routinely-collected health Data (RECORD) [16] guidelines were followed.

### Propensity score

Confounding by indication was adjusted for by using propensity scores in a Cox model [17]. As a sensitivity analysis to investigate the robustness of our models, propensity trimming on the 20th percentiles was performed to remove extreme propensity scores, i.e. patients with very high or very low probability of receiving a specific medication. For the within-class analysis the balance for baseline covariates was investigated using quintiles of the propensity scores. The strategy of inverse probability weighting (IPW) was also used in a Cox model as an extra tool to achieve homogeneity between the groups.

## Results

Baseline characteristics of this population-based cohort, comprising 22,476 HF patients followed over a 15-year period are presented in Table 1. The mean age was 76.8 years (SD 10.9 years), and 50.9% were women. The median number of medications prescribed at hospital discharge was 7 (IQR 5–10), after a median length of hospital stay of 6 days (IQR 3–11 days). Thirty percent of the patients was readmitted for HF. Median time to end of follow-up or readmission was 29.3 months (IQR 7.4–69.7 months).

### Main groups of heart failure medication

In Table 2A age, gender and number of medications are shown for each group of medications. Patients prescribed ACEI/ARB or BB are slightly younger compared to non-users, while patients using diuretics are older. Females are prescribed ACEI/ARB to a fewer extent, while diuretics are prescribed more often. To patients prescribed all groups of medication, a higher number of medications is prescribed, indicating more comorbidities.

Kaplan-Meier curves based on crude data for time to readmission are shown in Figs 1–6. Patients on MRA and diuretics have an increased risk of readmission (all *P*<0.001). The HR for readmission of ACEI and/or ARB and BB were not increased, in contrast to the HR of MRA and diuretics which were increased (Fig 7). All HRs shown are adjusted for age, gender, number of medications and year of admission.

**Table 1. Characteristics of the study cohort including 22476 patients.**

| | |
|---|---|
| Number of patients | 22476 |
| Age (years), mean (SD) | 76.8 (10.9) |
| Gender, female (%) | 11449 (50.9%) |
| Length of stay in days, median (interquartile range) | 6 (3–11) |
| Medication profile on discharge | |
| Number of medications, median (interquartile range) | 7 (5–10) |
| ACEI and/or ARB: | 14096 (62.7%) |
| ACEI (%) | 10599 (47.2%) |
| ARB (%) | 3898 (17.3%) |
| Beta-blocker: | 13406 (59.6%) |
| Metoprolol (%) | 6847 (30.5%) |
| Bisoprolol (%) | 3156 (14.0%) |
| Nebivolol (%) | 374 (1.7%) |
| Carvedilol (%) | 1224 (5.4%) |
| Mineralocorticoid-receptor antagonist: | 8317 (37.0%) |
| Spironolactone (%) | 7940 (35.3%) |
| Eplerenone (%) | 393 (1.7%) |
| Diuretic (%) | 18384 (81.8%) |
| Beta-blocker + ACEI/ARB (%) | 9088 (40.4%) |
| Beta-blocker + ACEI/ARB + MRA (%) | 3850 (17.1%) |

ACEI: angiotensin-converting enzyme inhibitor.

ARB: angiotensin receptor blocker.

SD: standard deviation.

## Within-class comparison

A within-class analysis was performed in patients using either ACEI or ARB (n = 13,695) and in patients using sBBHF or carvedilol (n = 11,541, Table 2B). Patients prescribed ACEI are younger and less often female compared to ARB. Patients prescribed carvedilol are considerably younger (71.6 years compared to 76.8 years) and less often female (40.8% vs 51.4%) compared to sBBHF.

Kaplan-Meier curves based on crude data for time to readmission are shown in Figs 1–6. Patients on carvedilol have an increased risk of readmission compared to patients on sBBHF (P<0.001).

After adjustment for age, gender, number of medications and year of admission the HRs for ARB relative to ACEI was 1.04 (95%CI 0.97–1.12), and for carvedilol relative to sBBHF 1.33 (95%CI 1.20–1.46) (Fig 7).

## Post-hoc analysis on time since discharge

In the post-hoc analysis (Table 3A) limiting the time of follow-up to 1, 6, 12 and 24 months, the HR shifted for ACEI and/or ARB from 1.01 (95%CI 0.96–1.06) with maximum follow-up, to 0.77 (95%CI 0.69–0.86) when the time since discharge was limited to 1 month. For BB the HR did not change when limiting the time of follow-up. The HR for MRA and diuretic shifted from above 1 with maximum follow-up to values below 1 when the time of follow-up was limited to one month.

When limiting the time of follow-up for the within-class comparison, the HRs did not change (Table 3A).

## Sensitivity analyses

The stability of the results was tested in various ways. Confounding by indication was adjusted for by using non weighted propensity scores in a Cox model but also inverse probability weighting was performed [17]. Furthermore, propensity trimming on the 20th percentiles was performed. These additional statistical analyses with propensity scores including among others other HF medication resulted in similar hazard ratios (S1 and S2 Files). For the within-class analysis the balance for baseline covariates was investigated using quintiles of the propensity scores and data were well balanced.

## Discussion

The favourable effects on HF hospital readmissions shown in RCTs for the four main groups of HF medication could only partly be reproduced in this observational study in a real-world,

**Table 2. Patient characteristics according to heart failure medication.**

**Table 2a Main groups of heart failure medication**

|  | Age (years), | Gender: | Number of medications: |
|---|---|---|---|
|  | mean (SD) | female (%) | median (IQR) |
| ACEI and/or ARB |  |  |  |
| yes | 76.0 (11.0) | 50.3 | 7 (5–9) |
| no | 78.1 (10.7) | 52.0 | 6 (4–9) |
| *P*-value | <0.001 | 0.019 | <0.001 |
| Beta-blocker |  |  |  |
| yes | 76.3 (11.0) | 51.2 | 7 (5–9) |
| no | 77.4 (10.7) | 50.5 | 6 (4–9) |
| *P*-value | <0.001 | 0.312 | <0.001 |
| Mineralocorticoid-receptor antagonist |  |  |  |
| yes | 76.8 (11.1) | 50.6 | 7 (5–10) |
| no | 76.7 (10.8) | 51.2 | 7 (4–9) |
| *P*-value | 0.754 | 0.383 | <0.001 |
| Diuretic |  |  |  |
| yes | 77.3 (10.8) | 51.9 | 7 (5–9) |
| no | 74.5 (11.1) | 46.6 | 5 (3–7) |
| *P*-value | <0.001 | <0.001 | <0.001 |

**Table 2b Within classes ACEI, ARB and ß-blockers**

|  | Age (years), | Gender: | Number of medications: |
|---|---|---|---|
|  | mean (SD) | female (%) | median (IQR) |
| ACEI versus ARB |  |  |  |
| ACEI | 75.7 | 48.4 | 8 (6–10) |
| ARB | 76.8 | 55.6 | 8 (6–11) |
| *P*-value | <0.001 | <0.001 | 0.001 |
| sBBHF versus carvedilol |  |  |  |
| sBBHF | 76.8 | 51.4 | 8 (6–11) |
| carvedilol | 71.6 | 40.8 | 8 (6–10) |
| *P*-value | <0.001 | <0.001 | 0.531 |

Number of patients in each group are shown in Tables 1 and 3.

For the main groups of heart failure medication the number of medications is calculated excluding the medication under surveillance.

ACEI: angiotensin-converting enzyme inhibitor.

ARB: angiotensin receptor blocker.

sBBHF: Selective β1-blocker with heart failure registration.

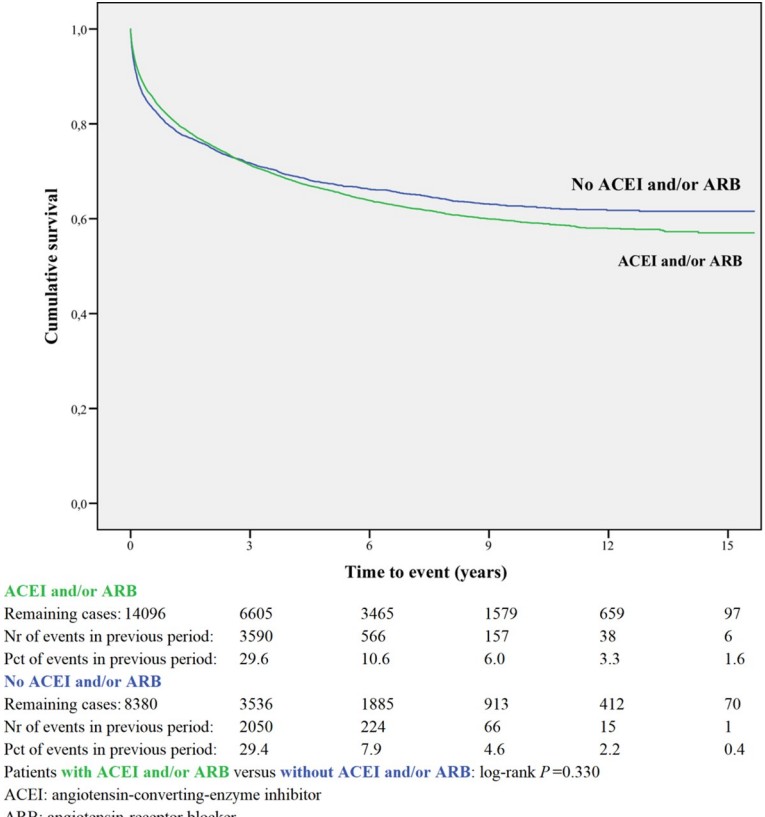

ACEI and/or ARB

| | | | | | |
|---|---|---|---|---|---|
| Remaining cases: 14096 | 6605 | 3465 | 1579 | 659 | 97 |
| Nr of events in previous period: | 3590 | 566 | 157 | 38 | 6 |
| Pct of events in previous period: | 29.6 | 10.6 | 6.0 | 3.3 | 1.6 |

No ACEI and/or ARB

| | | | | | |
|---|---|---|---|---|---|
| Remaining cases: 8380 | 3536 | 1885 | 913 | 412 | 70 |
| Nr of events in previous period: | 2050 | 224 | 66 | 15 | 1 |
| Pct of events in previous period: | 29.4 | 7.9 | 4.6 | 2.2 | 0.4 |

Patients **with ACEI and/or ARB** versus **without ACEI and/or ARB**: log-rank $P$ =0.330
ACEI: angiotensin-converting-enzyme inhibitor
ARB: angiotensin-receptor blocker

**Fig 1. Readmission for ACEI and/or ARB or non-use.**

large, unselected group of HF patients. After adjustment for age, gender, number of medications and year of admission the HR for being readmitted were equal for users versus non-users of ACEI and/or ARB and BB. MRA and diuretics slightly increased the risk of readmission relative to non-use. Patients using ACEI have the same risk of readmission compared to ARB, while patients prescribed carvedilol have an increased risk of readmission relative to patients on sBBHF. The post-hoc analysis showed that the protective effect shown for most HF medications shortly after discharge faded away when time since discharge increased. The additional statistical analyses resulted in similar hazard ratios.

## RCT versus real-world study

Remarkably, the results of our real-world observational study are different from those in RCTs. We will try to explain this. Randomizing the patients in RCTs beforehand between two or more interventions provides the greatest chance on equal prognosis of the patients at baseline and internal validity of the study. In an observational study like this the absence of randomization might cause confounding by indication or maybe more correctly: confounding by severity. However, the adjustment with propensity scores resulted in similar hazard ratios.

Where internal validity might be a challenge for observational studies, the external validity is a strong argument to choose this type of study design. Patients in RCTs are a careful selection of the population with HF: they are mostly younger, the percentage of men is higher and patients with multimorbidity are under-represented [1, 2, 18]. The number of medications (median 7) prescribed at hospital discharge in our HF patients, indicates a high number of

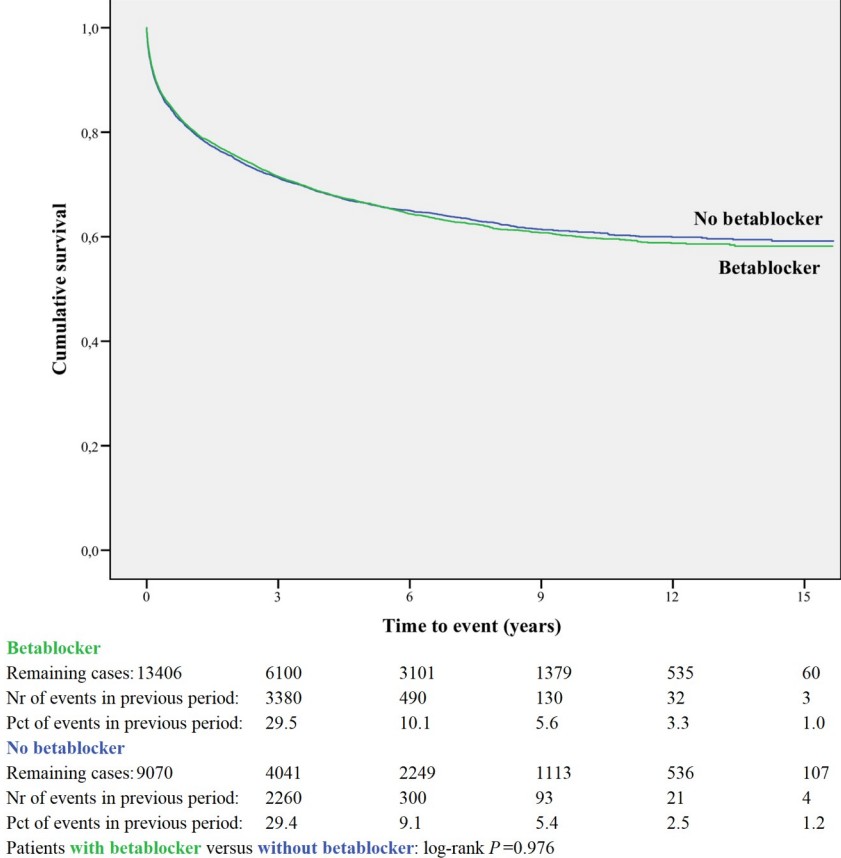

| Betablocker | | | | | |
|---|---|---|---|---|---|
| Remaining cases: 13406 | 6100 | 3101 | 1379 | 535 | 60 |
| Nr of events in previous period: 3380 | 490 | 130 | 32 | 3 | |
| Pct of events in previous period: 29.5 | 10.1 | 5.6 | 3.3 | 1.0 | |
| **No betablocker** | | | | | |
| Remaining cases: 9070 | 4041 | 2249 | 1113 | 536 | 107 |
| Nr of events in previous period: 2260 | 300 | 93 | 21 | 4 | |
| Pct of events in previous period: 29.4 | 9.1 | 5.4 | 2.5 | 1.2 | |

Patients **with betablocker** versus **without betablocker**: log-rank $P$ =0.976

**Fig 2. Readmission for betablocker or non-use.**

comorbidities inherent to the real-world scenario [19]. As we have cardiovascular as well as non-cardiovascular medication, we were able to combine these and use the number of comorbidities as an adjustment method.

Observational research in the format of a retrospective cohort study, based on data from validated electronic health records like this study, has the advantage to include many patients seen in daily practice, collected in a structured way [20–22]. Patient registries are often used in observational cardiology research, and have advantages and disadvantages compared to retrospective cohort studies. Registries are for example limited in the amount of data that can be collected for practical reasons, there is a risk for data quality issues and the external validity of registries is harder to guarantee. For example: clinical centers joining registry studies might be early adapters and active in optimizing contemporary HF treatment in general. Age and gender are straightforward parameters to compare the "real world"-degree of an RCT, registry or retrospective cohort study [2].

The database covered a 15-year period in which guidelines developed and prescribing patterns followed. The overall relatively low percentage of prescription of disease-modifying medications might reflect the fact that a substantial number of patients without these drugs might have HF with preserved ejection fraction, and consequently they could have a lower risk of HF hospitalizations. However, the percentages of patients to whom certain medications were prescribed, did not differ much from other studies, especially those in observational research as discussed in the earlier publication [13].

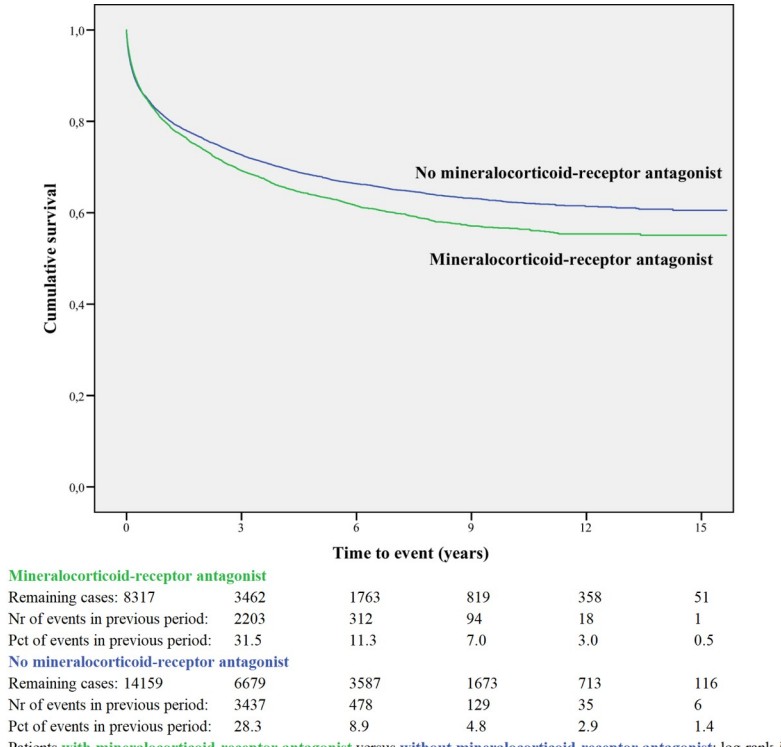

| Mineralocorticoid-receptor antagonist | | | | | |
|---|---|---|---|---|---|
| Remaining cases: 8317 | 3462 | 1763 | 819 | 358 | 51 |
| Nr of events in previous period: | 2203 | 312 | 94 | 18 | 1 |
| Pct of events in previous period: | 31.5 | 11.3 | 7.0 | 3.0 | 0.5 |
| No mineralocorticoid-receptor antagonist | | | | | |
| Remaining cases: 14159 | 6679 | 3587 | 1673 | 713 | 116 |
| Nr of events in previous period: | 3437 | 478 | 129 | 35 | 6 |
| Pct of events in previous period: | 28.3 | 8.9 | 4.8 | 2.9 | 1.4 |

Patients **with mineralocorticoid-receptor antagonist** versus **without mineralocorticoid-receptor antagonist**: log-rank *P* <0.001

**Fig 3. Readmission for mineralocorticoid-receptor antagonist or non-use.**

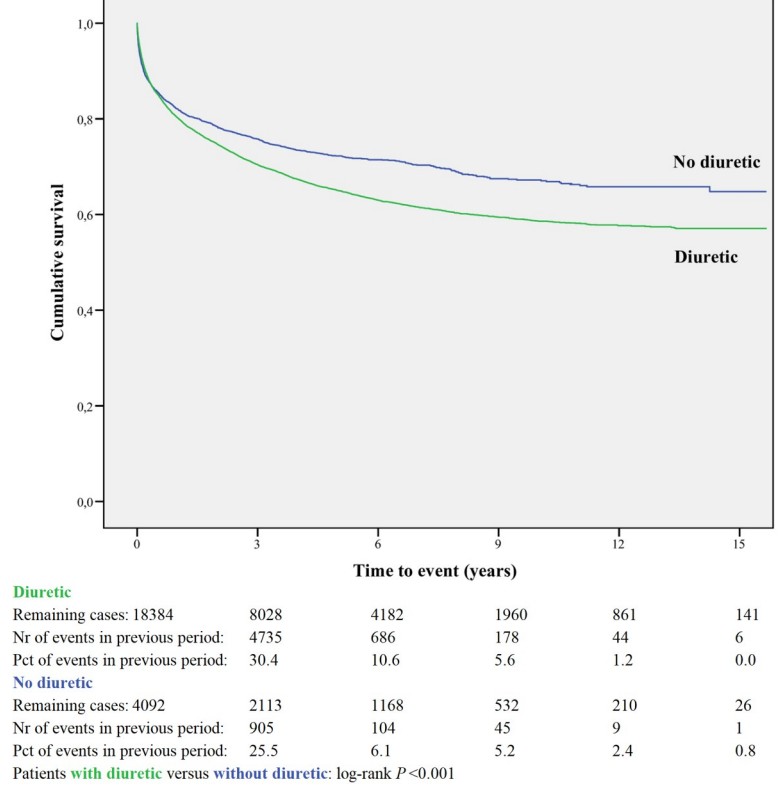

| Diuretic | | | | | |
|---|---|---|---|---|---|
| Remaining cases: 18384 | 8028 | 4182 | 1960 | 861 | 141 |
| Nr of events in previous period: | 4735 | 686 | 178 | 44 | 6 |
| Pct of events in previous period: | 30.4 | 10.6 | 5.6 | 1.2 | 0.0 |
| No diuretic | | | | | |
| Remaining cases: 4092 | 2113 | 1168 | 532 | 210 | 26 |
| Nr of events in previous period: | 905 | 104 | 45 | 9 | 1 |
| Pct of events in previous period: | 25.5 | 6.1 | 5.2 | 2.4 | 0.8 |

Patients **with diuretic** versus **without diuretic**: log-rank *P* <0.001

**Fig 4. Readmission for diuretic or non-use.**

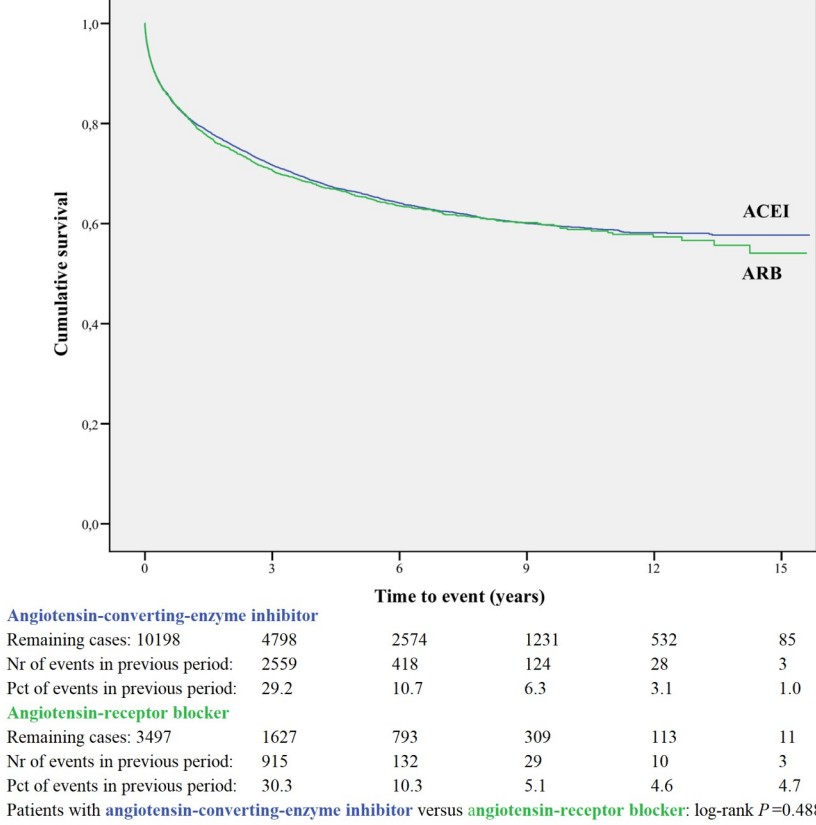

**Fig 5. Readmission for angiotensin-converting-enzyme inhibitor or angiotensin-receptor blocker.**

## Angiotensin-converting-enzyme inhibitors and angiotensin-receptor blockers

ACEI/ARB showed to be more effective in the period immediately following discharge, with gradually less effect up to 2 years. This was supported by a post hoc analysis of the Reality-AHF registry, which showed that prescription of ACEI/ARB or BB before the time of discharge was not associated with reduced HF readmission after one year in patients hospitalized with HFrEF [23]. In a registry study [7] discharge use of ACEI or ARB were on the list of the most important predictors for the combined end point of death or readmission in the first 60 to 90 days after discharge, risks were reduced when using ACEI or ARB. In Medicare beneficiaries (a retrospective cohort study) the use of ACEI/ARB was also associated with a lower risk of 30-day all-cause readmission and remained lower at 1 year post discharge among admitted patients with HFrEF [24].

## Beta-blockers

When all four BBs are combined, we observed a somewhat lower risk for readmission in the short term without any effect in the long-term. In contrast, carvedilol users had higher risks of readmission, independent of time since discharge. Various studies are published which -in general- show a moderately reduced readmission risk for BBs compared to non-users [25–30].

A within-class analysis was performed between patients using sBBHF or carvedilol. The HR for carvedilol relative to sBBHF was increased and did not change over time. According to the

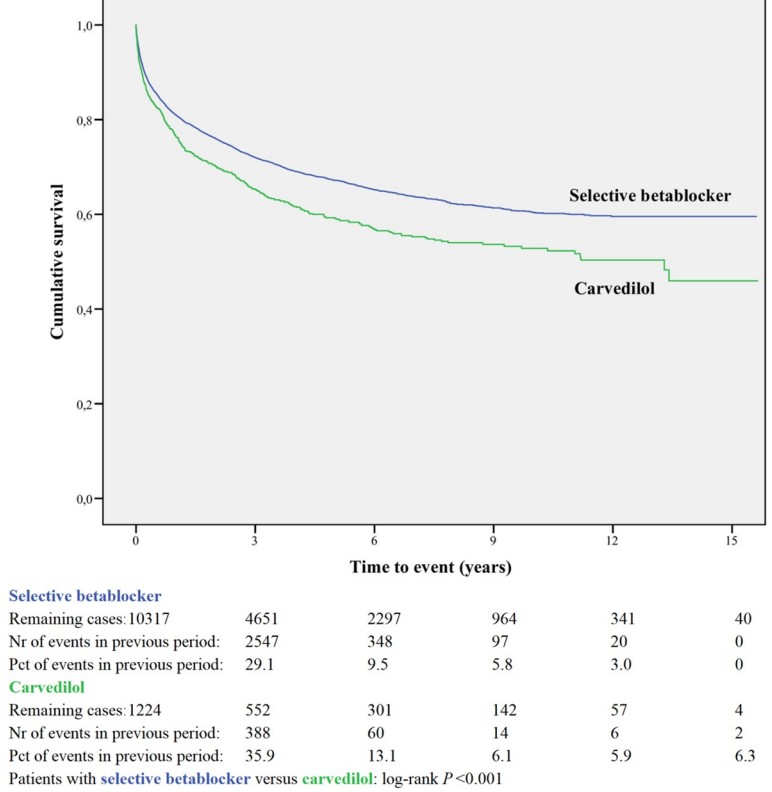

**Selective betablocker**

| | | | | | |
|---|---|---|---|---|---|
| Remaining cases: 10317 | 4651 | 2297 | 964 | 341 | 40 |
| Nr of events in previous period: 2547 | 348 | 97 | 20 | 0 | |
| Pct of events in previous period: 29.1 | 9.5 | 5.8 | 3.0 | 0 | |

**Carvedilol**

| | | | | | |
|---|---|---|---|---|---|
| Remaining cases: 1224 | 552 | 301 | 142 | 57 | 4 |
| Nr of events in previous period: 388 | 60 | 14 | 6 | 2 | |
| Pct of events in previous period: 35.9 | 13.1 | 6.1 | 5.9 | 6.3 | |

Patients with **selective betablocker** versus **carvedilol**: log-rank *P* <0.001

**Fig 6. Readmission for selective betablocker or carvedilol.**

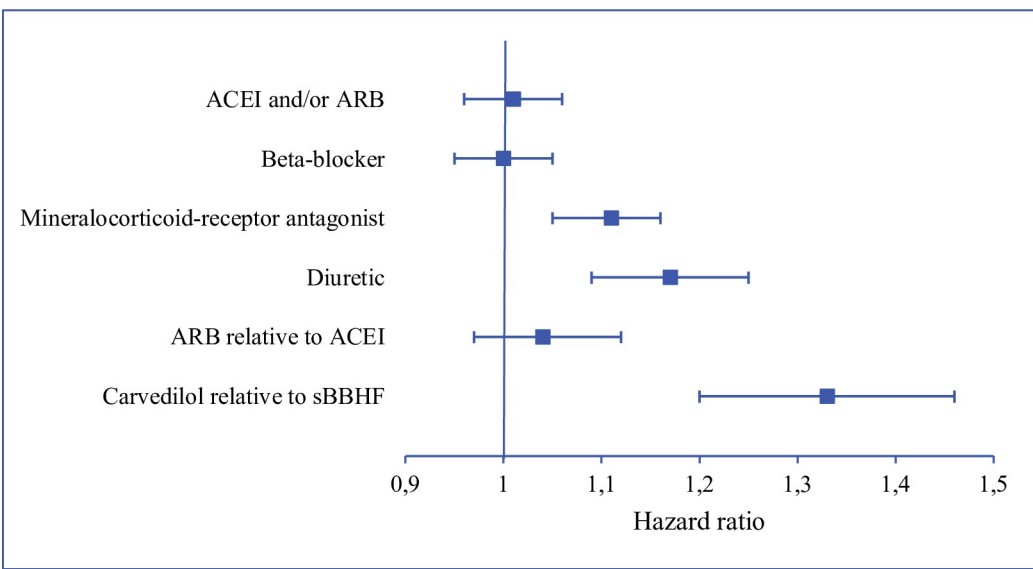

**Fig 7. Risks of heart failure readmission.** ACEI and/or ARB, beta-blocker, mineralocorticoid-receptor antagonist and diuretic: use of medication relative to non-use. sBBHF: β1-selective beta-blocker with a registration for heart failure. ACEI: angiotensin-converting enzyme inhibitor. ARB: angiotensin receptor blocker.

**Table 3. Risk of readmission with varying time since discharge.**

| Table 3a Hazard ratio* for heart failure medication relative to non-use | | | | | | |
|---|---|---|---|---|---|---|
| Time since discharge limited to: | n: | 1 month | 6 months | 12 months | 24 months | maximum follow-up |
| ACEI and/or ARB | 14096 | 0.77 | 0.82 | 0.87 | 0.93 | 1.01 |
| | | 0.69–0.86 | 0.77–0.88 | 0.82–0.93 | 0.88–0.99 | 0.96–1.06 |
| ACEI | 10198 | 0.78 | 0.82 | 0.87 | 0.92 | 1.00 |
| | | 0.69–0.88 | 0.76–0.88 | 0.82–0.93 | 0.86–0.98 | 0.95–1.05 |
| ARB | 3497 | 0.75 | 0.83 | 0.88 | 0.96 | 1.04 |
| | | 0.63–0.88 | 0.75–0.93 | 0.80–0.97 | 0.89–1.05 | 0.97–1.12 |
| Beta-blocker | 13406 | 0.93 | 0.94 | 0.97 | 0.96 | 1.00 |
| | | 0.83–1.04 | 0.87–1.01 | 0.91–1.03 | 0.90–1.01 | 0.95–1.05 |
| sBBHF | 10317 | 0.90 | 0.95 | 0.97 | 0.96 | 0.99 |
| | | 0.81–1.01 | 0.88–1.02 | 0.91–1.03 | 0.90–1.02 | 0.94–1.04 |
| Carvedilol | 1224 | 1.21 | 1.15 | 1.23 | 1.24 | 1.28 |
| | | 0.98–1.51 | 0.99–1.33 | 1.08–1.39 | 1.11–1.40 | 1.16–1.41 |
| Mineralocorticoid-receptor antagonist | 8317 | 0.88 | 0.99 | 1.02 | 1.06 | 1.11 |
| | | 0.79–0.99 | 0.92–1.06 | 0.96–1.08 | 1.00–1.12 | 1.05–1.16 |
| Diuretic | 18384 | 0.76 | 0.96 | 1.01 | 1.07 | 1.17 |
| | | 0.67–0.87 | 0.88–1.06 | 0.93–1.10 | 0.99–1.15 | 1.09–1.25 |
| Table 3b Hazard ratio* for the analysis of ACEI versus ARB and between ß-blockers | | | | | | |
| Time since discharge limited to: | | 1 month | 6 months | 12 months | 24 months | maximum follow-up |
| ARB relative to ACEI | 13695 | 0.94 | 1.00 | 0.99 | 1.04 | 1.04 |
| | | 0.79–1.10 | 0.90–1.11 | 0.91–1.09 | 0.96–1.13 | 0.97–1.12 |
| Carvedilol relative to sBBHF | 11541 | 1.38 | 1.25 | 1.29 | 1.33 | 1.33 |
| | | 1.11–1.72 | 1.08–1.45 | 1.14–1.47 | 1.18–1.49 | 1.20–1.46 |

sBBHF: β1-selective beta-blocker with a registration for heart failure.

ACEI: angiotensin-converting enzyme inhibitor.

ARB: angiotensin receptor blocker.

*with 95% confidence interval.

n: number of patients.

ATC classification of the WHO, based on pharmacological mechanism of action, three of the BB are β1-selective BB (bisoprolol, metoprolol and nebivolol), while carvedilol is classified as an α- and β-blocking agent. The hypothesis is that carvedilol might have an efficacy advantage in HF because of blockade of additional myocardial β2- and (partly) α1-receptors [26, 31]. However, the β1-receptor is selectively downregulated in the failing heart, whereas β2-receptors are unchanged in protein abundance and α1-receptors are actually increased.

Concerning the α1-blockade, the Allhat-study [32] showed that use of α-receptor blockers was associated with a 25% increased risk of combined cardiovascular disease events, including twice the risk of developing congestive HF. In the Veterans Administration Cooperative Study [33], HF patients receiving the α1-blocker prazosin experienced worse outcomes than did those receiving the combined vasodilator therapy of hydralazine and isosorbide dinitrate. This negative effect of α1-blockade might be a plausible explanation for the increased risk of readmission we found for carvedilol. α1-Blockers are, after all, listed as "drugs that may cause or exacerbate HF" [34].

Concerning the β-receptor blocking effect, the only RCT is the Comet trial comparing metoprolol and carvedilol [35]. The authors suggested that the non-selective BB carvedilol extends survival compared with the selective BB metoprolol, with a composite endpoint of mortality or

all-cause admission occurring in 1116 (74%) of 1511 on carvedilol and in 1160 (76%) of 1518 on metoprolol (HR 0.94; 95%CI 0.86–1.02). However, the postulated added effect of β2-blockade has caused discussion [31, 36, 37]. First, the fact that the negative effect of BB on the myocardium is solely caused by β1-blockade, as described above. Secondly, it was pointed out that molecular effects of carvedilol and metoprolol are similar when equipotent doses are used. Thirdly, the fact that there were no supportive outcome data from large clinical trials in HF for benefit of α1-blockade. In several observational studies carvedilol was compared to sBBHF, especially metoprolol. Two registry studies showed that the effectiveness of carvedilol and metoprolol in patients with HF is similar [26, 38]. Three retrospective cohort studies have been published. Two of them [39, 40] showed no difference on readmission or survival. Only the database-study of Bølling [41], starting in 1978, showed that HF patients receiving high-dose carvedilol (≥50 mg daily) showed significantly lower all-cause mortality risk and hospital admission risk, compared with other BB. Age and gender distribution in the retrospective cohort studies were comparable to our study, patients in both registries were younger and the percentage of men is higher. Therefore from observational studies there is no proof that carvedilol results in lower readmission risk than sBBHF, while the only RCT is criticized.

Carvedilol is presumably more often prescribed to patients with more severe HF, related to the placebo-controlled Copernicus-study [42] in which patients with moderate to severe HFrEF were included. This study was the major RCT for market authorisation of carvedilol. Although we did try to adjust for confounding by indication by using propensity scores, probably residual confounding is still present. Furthermore, there might be subgroups of patients, for whom carvedilol is a more appropriate BB. However, our results suggest that patient prescribed carvedilol have a higher or at least similar risk of HF readmission compared to sBBHF.

## Mineralocorticoid-receptor antagonists

MRA were associated with an increased risk for readmission compared to non-users after 2 years. However, MRA showed to be effective in the first month immediately following discharge. In the Rales study [43], the major RCT for MRA, treatment with spironolactone led to a relative risk reduction in HF admission of 35% within an average of 2 years since starting treatment. A retrospective cohort study immediately after the publication of RALES recorded abrupt increases in the rate of prescriptions for spironolactone and in hyperkalemia-associated morbidity and mortality [44]. Patients in the Rales study were much younger (65 years versus 78 years) and less often female (27% versus 50%), demonstrating the differences between an RCT and an observational study. An attempt was made to replicate the Rales study using a propensity score matching approach, with the ultimate objective of bridging from the trial population to a real world population of people with HF [45]. Survival in the spironolactone group in the Rales study and in the propensity matched study were remarkably similar, with just over 80% survival in both cohorts at one year. However, when they compared the tightly matched propensity score groups, rather than reducing mortality, spironolactone seemed to be associated with a substantial increase in the risk of death supposedly caused by confounding by indication. These studies on MRA illustrate the difficulty of reconfirming results from an RCT in observational research.

## Diuretics

Loop diuretics are recommended in chronic heart failure to prevent signs and symptoms of congestion [46]. However, it is disputed whether diuretics are disease severity markers or true risk factors [47]. In our study, the risk of HF readmission was high (HR 1.17). However, when we limited the time since discharge in a post-hoc analysis, diuretics showed to be effective in

the period immediately following discharge. That might be explained by the occurrence of electrolyte disturbances, further neurohormonal activation, accelerated kidney function decline, and symptomatic hypotension. Once euvolaemia has been achieved, loop diuretic therapy should be continued at the lowest dose that can maintain euvolaemia. In the most congested patients diuretics might not be able to fully treat congestion, leading to HF readmission.

### Strengths and limitations

A major strength of our study is the large number of patients and the fact that not only the cardiovascular medication is available, but non-cardiovascular medication as well, which we used as a proxy for the number of comorbidities. The increasing number of comorbidities [48] as well as the increase in the age of the HF patients make it more difficult to translate the results of carefully designed RCTs to daily clinical practise, due to the inclusion of selective groups of patients. Observational research can be used to close this gap.

We have shown the effect of time since discharge. To the best of our knowledge, this has not been done before. However, a limitation of the study was the fact that we have in our data extraction limited information about the period between the discharge and the end of follow-up: long-term data on post-discharge medication use (e.g. adherence) or the development of comorbidities. A part of the patients will still be in the start-up phase of disease modifying therapy, because it is their first hospital admission for HF. Because of the importance of starting HF medication during admission, we have chosen to investigate the medication profile at discharge and not to include new prescriptions once a patient had been discharged. Furthermore, information about the severity of HF of the patient is not available. Patients who have died, were not at risk anymore for readmission. As we have no information on the cause of the end of follow-up, this implies that it was not possible to correct for this competing risk.

Although we did try to adjust for confounding by indication by using propensity scores as a variable in the Cox model and with IPW [17], residual, unmeasured confounding might still be present, especially as we calculated propensity scores using logistic regression analyses based on limited information of the patient and no information on HF aetiology, comorbidities, left ventricular ejection fraction or functional class (NYHA). Therefore, it was not possible to investigate for example the association between the ejection fraction and readmission. The database does not contain this information, which would have made the conclusions more robust. Also, less residual confounding by indication would have been present in the within-class comparison and the post-hoc analysis on time since discharge as confounders are equally present.

### Conclusion

This observational database study investigated the association between HF medication and readmission for HF in daily clinical practise in a group of patients discharged from hospital after a first admission for HF. Patients prescribed carvedilol have a higher or at least a similar risk of HF readmission compared to sBBHF, an observation which should be confirmed in an RCT. This study showed that all groups of HF medication -some more pronounced than others- were more effective immediately following discharge than with a long follow-up.

### Supporting information

**S1 File. Propensity scores within main groups of heart failure medication.**
(PDF)

**S2 File. Propensity scores for ACEI versus ARB and between β-blockers.**
(PDF)

## Acknowledgments

The authors would like to thank all healthcare providers contributing information to the PHARMO Database Network.

## Author Contributions

**Conceptualization:** Willemien J. Kruik-Kollöffel, Job van der Palen, Carine J. M. Doggen, Marissa C. van Maaren, H. Joost Kruik, Edith M. Heintjes, Kris L. L. Movig, Gerard C. M. Linssen.

**Data curation:** Willemien J. Kruik-Kollöffel, Carine J. M. Doggen, Edith M. Heintjes, Gerard C. M. Linssen.

**Formal analysis:** Willemien J. Kruik-Kollöffel, Job van der Palen, Marissa C. van Maaren, H. Joost Kruik, Edith M. Heintjes, Kris L. L. Movig, Gerard C. M. Linssen.

**Investigation:** Willemien J. Kruik-Kollöffel, Job van der Palen, Carine J. M. Doggen, Marissa C. van Maaren, H. Joost Kruik, Edith M. Heintjes, Kris L. L. Movig, Gerard C. M. Linssen.

**Methodology:** Willemien J. Kruik-Kollöffel, Job van der Palen, Carine J. M. Doggen, Marissa C. van Maaren, Edith M. Heintjes, Gerard C. M. Linssen.

**Project administration:** Willemien J. Kruik-Kollöffel.

**Software:** Willemien J. Kruik-Kollöffel, Carine J. M. Doggen, Marissa C. van Maaren, Edith M. Heintjes.

**Supervision:** Willemien J. Kruik-Kollöffel, Job van der Palen, Gerard C. M. Linssen.

**Validation:** Willemien J. Kruik-Kollöffel, Job van der Palen, Carine J. M. Doggen, Edith M. Heintjes.

**Visualization:** Willemien J. Kruik-Kollöffel, Job van der Palen, Gerard C. M. Linssen.

**Writing – original draft:** Willemien J. Kruik-Kollöffel, Job van der Palen, Carine J. M. Doggen, Marissa C. van Maaren, H. Joost Kruik, Edith M. Heintjes, Kris L. L. Movig, Gerard C. M. Linssen.

**Writing – review & editing:** Willemien J. Kruik-Kollöffel, Job van der Palen, Carine J. M. Doggen, Marissa C. van Maaren, H. Joost Kruik, Edith M. Heintjes, Kris L. L. Movig, Gerard C. M. Linssen.

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
