## [Decision Letter · Decision Letter 0]

21 Oct 2020

PONE-D-20-26501

Heart failure medication after a first hospital admission and risk of heart failure readmission, focus on beta-blockers and renin-angiotensin-aldosterone system medication: A retrospective cohort study in linked databases

PLOS ONE

Dear Dr. Kruik-Kollöffel,

Thank you for submitting your manuscript to PLOS ONE. After careful consideration, we feel that it has merit but does not fully meet PLOS ONE’s publication criteria as it currently stands. Therefore, we invite you to submit a revised version of the manuscript that addresses the points raised during the review process.

We look forward to receiving your revised manuscript.

Kind regards,

Antonio Cannatà

Academic Editor

PLOS ONE

Journal Requirements:

2.We note that you have indicated that data from this study are available upon request. PLOS only allows data to be available upon request if there are legal or ethical restrictions on sharing data publicly. For information on unacceptable data access restrictions, please see http://journals.plos.org/plosone/s/data-availability#loc-unacceptable-data-access-restrictions.

Reviewers' comments:

Reviewer's Responses to Questions

**Comments to the Author**

1. Is the manuscript technically sound, and do the data support the conclusions?

Reviewer #1: Partly

Reviewer #2: Yes

2. Has the statistical analysis been performed appropriately and rigorously? 

Reviewer #1: Yes

Reviewer #2: Yes

3. Have the authors made all data underlying the findings in their manuscript fully available?

Reviewer #1: Yes

Reviewer #2: Yes

4. Is the manuscript presented in an intelligible fashion and written in standard English?

Reviewer #1: Yes

Reviewer #2: Yes

5. Review Comments to the Author

Reviewer #1: In the manuscript written by Kruik-Kollöffel W. et al. the authors analyzed the relationship between HF medications prescription at discharge and the risk oh heart hospitalization during follow up in a large unselected HF population. The manuscript gives us some interesting insights: beta blockers and ACE-inhibitors seem to confer a protective effect in the first months after hospitalization, but lately this favourable effect might disappear. Carvedilol compared to other selective beta-blockers seems associated to a higher risk profile. Diuretics and mineralcorticoid-receptor antagonists seem associated with a higher risk of hospitalizations.

These results are interesting, showing us that the effect of HF medications might be less convincent in unselected populations from observational registries than in randomized clinical trials.

However some important limitations should be mentioned:

1) As the authors stated in the limitations section, lack of characterization of the presented population is a strong limitation. Heart failure is an extreme heterogeneous disease, and the efficacy drug therapy is strongly affected by the type of heart failure and etiology.

2) The relatively low percentage of ACE-inhibitors, beta-blockers and mineralcorticoid-receptor antagonists prescription might reflect the fact that a substantial number of patients without these drugs might had HFpEF, and consequently they could have a lower risk oh HF hospitalizations. This might surely be a confounding factor that attenuated the expected benefit of HF drugs and it should be better discussed by the authors.

3) About the comorbidities: do the authors have data about the renal function? I might expect that the possible harfmul effect of MRA and diuretics is more evident in patients with chronic kidney disease where a strict follow up of these drugs should be done.

4) I suggest to add the percentage of the events in the figure 1 to 6 in addition to the number of events.

Reviewer #2: Thank you for reviewing your paper and addressing the comments of the reviewers and of the editor. In my opinion, the quality of the paper has improved and it is more informative than the previous version.

6. PLOS authors have the option to publish the peer review history of their article (what does this mean?). If published, this will include your full peer review and any attached files.

Reviewer #1: **Yes: **Paolo Manca

Reviewer #2: No

---

## [Editor Report · Decision Letter 1]

7 Dec 2020

Heart failure medication after a first hospital admission and risk of heart failure readmission, focus on beta-blockers and renin-angiotensin-aldosterone system medication: A retrospective cohort study in linked databases

PONE-D-20-26501R1

Dear Dr. Kruik-Kollöffel,

We’re pleased to inform you that your manuscript has been judged scientifically suitable for publication and will be formally accepted for publication once it meets all outstanding technical requirements.

Kind regards,

Antonio Cannatà

Academic Editor

PLOS ONE

Additional Editor Comments (optional):

Please consider improving the quality of the images inserting the patients at risk below the KM curves and optimising the KM curves to maximise their impact.

---

## [Editor Report · Acceptance letter]

11 Dec 2020

PONE-D-20-26501R1 

Heart failure medication after a first hospital admission and risk of heart failure readmission, focus on beta-blockers and renin-angiotensin-aldosterone system medication: A retrospective cohort study in linked databases 

Dear Dr. Kruik-Kollöffel:

I'm pleased to inform you that your manuscript has been deemed suitable for publication in PLOS ONE. Congratulations! Your manuscript is now with our production department. 

Kind regards, 

on behalf of

Dr. Antonio Cannatà 

Academic Editor

PLOS ONE